

# Indicators of distress in newly diagnosed breast cancer patients

Andrea Chirico[1,2], Fabio Lucidi[2], Luca Mallia[2], Massimiliano D'Aiuto[1] and Thomas V. Merluzzi[3]

[1] Breast Cancer Department, National Cancer Institute Fondazione "G.Pascale," Naples, Italy
[2] Department of Psychology of Developmental and Socialisation Processes, "Sapienza" University of Rome, Rome, Italy
[3] Department of Psychology, Notre Dame University, Notre Dame, USA

## ABSTRACT

**Background.** The diagnosis, treatment, and long-term management of cancer can present individuals with a multitude of stressors at various points in that trajectory. Psychosocial distress may appear early in the diagnostic process and have negative effects on compliance with treatment and subsequent quality of life.

**Purpose.** The aim of the study was to determine early-phase predictors of distress before any medical treatment.

**Method.** Consistent with the goals of the study, 123 newly diagnosed breast cancer patients (20 to 74 years old) completed multiple indicators of knowledge about breast cancer management and treatment, attitudes toward cancer, social support, coping efficacy, and distress.

**Results.** SEM analysis confirmed the hypothesized model. Age was negatively associated with the patient's knowledge ($\beta = -0.22$), which, in turn, was positively associated with both attitudes toward breast cancer ($\beta = 0.39$) and coping self-efficacy ($\beta = 0.36$). Self-efficacy was then directly related to psychological distress ($\beta = -0.68$).

**Conclusions.** These findings establish indicators of distress in patients early in the cancer trajectory. From a practical perspective, our results have implications for screening for distress and for the development of early interventions that may be followed by healthcare professionals to reduce psychological distress.

Corresponding author
Andrea Chirico,
andrea.chirico@uniroma1.it,
dr.andreachirico@gmail.com

## INTRODUCTION

Cancer is the second leading cause of death among women in Italy, and breast cancer is the most prevalent type accounting for 17.1% of all cancer deaths in women each year. In all phases of the cancer trajectory, from diagnosis and treatment to long-term management, patients may experience financial strains, difficulty in interpersonal relationships, physical symptoms, and emotional distress (*Philip et al., 2013*). The prevalence of psychological distress among breast cancer patients is higher than the general population, which increases the risk for developing clinical levels of anxiety and depression (*Burgess et al., 2005*; *Deshields et al., 2006*; *Mehnert & Koch, 2008*; *Vahdaninia, Omidvari & Montazeri,*

*2010; Montgomery & McCrone, 2010; Hill et al., 2011*) that can adversely affect treatment compliance.

Whereas there is a great deal of literature on distress during the course of treatment (*Lepore & Coyne, 2006*), less is known about the time between diagnosis and the beginning of treatment. Although, research demonstrates that moderate to high levels of psychosocial distress appear early on in the cancer diagnosis process (e.g., *Nosarti et al., 2001; Lauzier et al., 2010; Andreu et al., 2012; Costa-Requena, Rodríguez & Fernández-Ortega, 2013*), it is important to also determine the demographic, social, and psychological variables that mitigate or lessen that initial distress, which then might set the course for coping with the disease and its treatments.

The current study focuses on the time after diagnosis, before treatment, and is imbedded in the biobehavioral model of cancer stress and disease course (*Andersen, Kiecolt-Glaser & Glaser, 1994*). Based on the biobehavioral model and the self-regulation processes in which people engage (*Carver & Scheier, 2000*), the early stages of the cancer trajectory may be critical in setting the course for reducing risk for clinical distress (*Lam et al., 2012*). In fact, the literature is rather clear on the relationship between distress and a number of issues that impinge upon engagement in treatment, recovery from illness, satisfaction with the provision of health care services (*Costanzo et al., 2007; Manning & Bettencourt, 2011*) and adjusting to life after treatment (*Burgess et al., 2005; Fiszer et al., 2014*). Congruent with this model several studies have revealed that a high level of preoperative or immediate postoperative distress (*Nosarti et al., 2001; Gallagher, Parle & Cairns, 2002; Badger et al., 2004; Lam et al., 2007; Millar et al., 2005; Lam et al., 2012*) resulted in poorer psychological outcomes in the subsequent treatment period than low levels of distress. Moreover, psychological distress had a negative effect on patients' quality of life, and, as noted earlier, on compliance with treatment (*Ayres et al., 1994; Colleoni et al., 2000; Bui et al., 2005; Reich, Lesur & Perdrizet-Chevallier, 2008; Manning & Bettencourt, 2011; Costa-Requena, Rodríguez & Fernández-Ortega, 2013; Philip et al., 2013*). Patients who are less anxious and depressed as they enter the treatment course of their cancer show a better adjustment to illness, request lower levels of medical attention and create lower medical costs than patients with higher levels of anxiety and depression (e.g., *Butler et al., 2006*).

Thus, early intervention may be the key to mitigating distress (*Casellas-Grau, Font & Vives, 2014*); however, a first step in that process would be to identify critical demographic, social, and psychological predictors of distress that may be the focus of that intervention.

As regards demographic variables, generally studies have not supported any relationship between patients' marital status or education and psychological distress (*Avis, Crawford & Manuel, 2005; Reich, Lesur & Perdrizet-Chevallier, 2008; Vahdaninia, Omidvari & Montazeri, 2010; Mertz et al., 2012*), but do report significant age differences in patients' psychological distress with younger age related to greater distress and poorer psychological adjustment following diagnosis compared to older age (*Van't Spijker, Trijsburg & Duivenvoorden, 1997; Avis et al., 2012; Mertz et al., 2012*). Yet, the effects of age are not uniformly related to distress (*Maunsell, Brisson & Deschi'nes, 1992; Philip et al., 2013*). Despite these differences in findings, a comprehensive analysis of age and adjustment

to cancer (*Mosher & Danoff-Burg, 2005*) stressed the importance of focusing on the relationship between age and patients' psychological distress and strongly recommended analyzing mediators of this relationship. Also, there is little known about age effects early on in the cancer trajectory.

Several studies and reviews focused their attention on social support in cancer patients in treatment (*Grassi et al., 1993*; *Merluzzi & Sanchez, 1997*; *Merluzzi et al., 2001*; *Friedman et al., 2006*; *Arora et al., 2007*; *Nausheen et al., 2009*; *Henselmans et al., 2010*; *Heitzmann et al., 2011*; *Philip et al., 2013*). Perceived social support (i.e., from family, friends and significant others) has been established as protective factor, which mitigates psychological distress in breast cancer patients (*Friedman et al., 2006*) and specifically in newly diagnosed patients (*Arora et al., 2007*; *Drageset et al., 2012*), and, therefore, is included in the model in the current study.

There is also evidence that coping mitigates or exacerbates distress over time in cancer patients by engaging several mechanisms. For example, disengagement and denial coping tend to undo the positive effects of optimism on distress in a mediated model of adjustment to breast cancer (*Carver et al., 1993*). Also, in a longitudinal model, emotionally expressive coping in breast cancer patients was associated with an increase in physical health and reductions in distress (*Stanton et al., 2000*). Along those lines, self–efficacy for coping, that is expectations about the ability to cope with cancer, plays a critical role in influencing cancer-related outcomes including distress. There is a negative relationship between perceived self-efficacy for coping with cancer and psychological distress in cancer patients (*Merluzzi & Sanchez, 1997*; *Merluzzi et al., 2001*; *Howsepian & Merluzzi, 2009*; *Heitzmann et al., 2011*; *Philip et al., 2013*) and, specifically, in breast cancer patients (*Henselmans et al., 2010*). Interestingly, self-efficacy for coping represents how the patient might expect to cope with cancer, and can be assessed even if the patient is not yet in treatment. Thus, it is a very relevant variable to assess for those newly diagnosed.

According to social cognitive theory (*Bandura, 1997*), self-efficacy is influenced by the personal knowledge and prior experiences (*Avci, 2008*; *Heitzmann et al., 2011*). Accordingly, several studies have established that women's personal knowledge about breast cancer, including its management and medical treatment, is inversely related to their psychological distress (*Ohaeri, Ofi & Campbell, 2012*), compliance with preventative behaviors such as mammography (*Holt, Lukwago & Kreuter, 2003*), and time orientation regarding the consequences of breast cancer (*Lukwago et al., 2003*). Thus, with regard to the proposed model, and consistent with Self-Efficacy Theory, coping self-efficacy would be expected to mediate the relationship between knowledge and psychological distress.

Though not as extensively studied as knowledge and coping efficacy, patients' attitudes towards cancer may be a precursor to coping. Most of the research on attitudes is in the context of health care providers' (*Johnson et al., 2013*) or people's attitudes (*Schernhammer et al., 2010*) toward prevention and screening. However, one study has shown that compared to positive attitudes, negative attitudes toward breast cancer and its treatment were associated with a higher level of psychological distress (*Gilbar, 2003*).

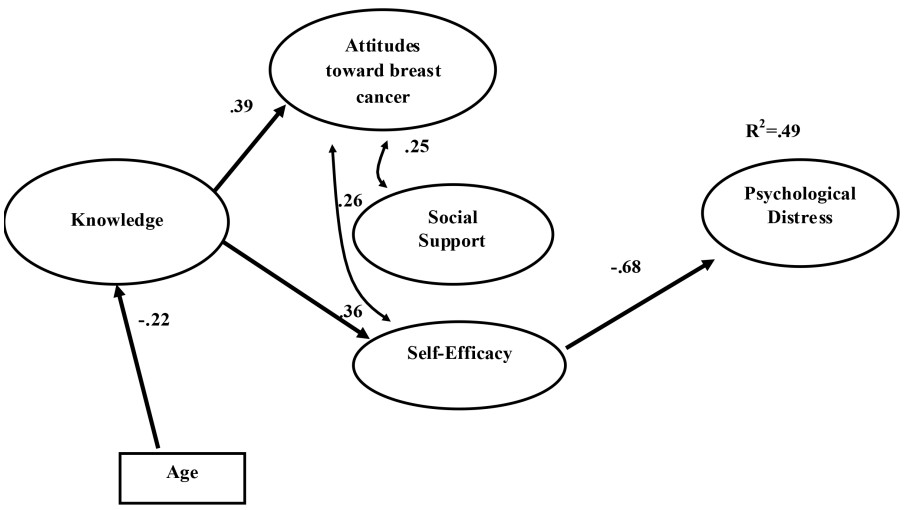

Paths significant at p<0.05

**Figure 1  Model.** Hypothesized model with estimation path. Path significant at $p < 0.05$.

In sum, in order to understand what might prevent distress early in the cancer trajectory, studies need to be conducted on the time between diagnosis and the beginning of treatment in which relevant variables are investigated in terms of their role in mitigating or exacerbating distress. Among those there is support for including age, knowledge, attitudes, social support and self-efficacy expectations for coping. In a cross-sectional design we tested a model in which we hypothesized that age is related to patient's knowledge about breast cancer, which in turn, is related to both attitudes toward breast-cancer treatment and coping self-efficacy. We further hypothesized that these latter two variables would directly related to the level of patients' psychological distress, and mediate the effects of knowledge. The independent contribution of social support was also evaluated (Fig. 1).

## MATERIALS & METHODS

### Participants

This study was conducted at the Breast Oncology Department of the National Cancer Institute 'Giovanni Pascale' Foundation in Naples, Italy. Medical consultants identified 130 newly diagnosed breast cancer patients during the period from January to April 2010. All patients were recruited during their hospitalization. These patients were admitted to hospital for examination and were then scheduled for surgery within 1–3 days. At the time of admission, the specific type of surgery that would be performed (e.g., Lumpectomy, Quadrantectomy, or Mastectomy) had not been determined. Also none of these patients had previously received adjuvant chemotherapy or any other cancer treatments (i.e., surgery or radiotherapy). Demographic information, staging data, and familial history of breast cancer were collected from medical records after obtaining informed consent and are contained in Table 1. All patients ($N = 130$) consented to

**Table 1** Description of the sample.

| Age distribution in percentiles | |
|---|---|
| 25th | 39.00 yrs |
| 50th | 45.00 yrs |
| 75th | 52.00 yrs |
| **Breast cancer stage** | |
| T1 | 55.9% |
| T2 | 20.5% |
| T3 | 14.2% |
| T4 | 9.4% |
| **Familiar history of breast cancer** | 32.8% |
| **Surgical treatment** | |
| Lumpectomy | 26.0% |
| Quadrantectomy with LNS | 51.2% |
| Mastectomy | 22.8% |

be interviewed for the study; however, seven did not answer all the interview questions and were excluded from the analysis. In the final sample, age ranged from 20 to 74 years ($M$ age $= 45.69$; $SD = 10.01$).

## Procedures

All the patients were approached by a psychologist who described the research project and presented the consent form. After giving consent they participated in a structured face-to-face interview lasting about 30 min administered by the first author (a psychologist trained in conducting diagnostic interviews).

## Measures

The interviews were based on the administration of the following measures.

### Knowledge about mammography and breast cancer

Knowledge about mammography, breast cancer, and breast cancer treatment was evaluated through scales successfully used in previous studies in prevention settings (*Holt et al., 2003*; *Holt, Lukwago & Kreuter, 2003*) and translated into Italian for the present study. In particular we evaluated: (1) *Mammography knowledge*: five items assessed perceptions of what mammograms can accomplish. For example, one item asked whether participants thought that having a mammogram could reduce their risk of dying from breast cancer. (2) *Breast cancer knowledge*: five items assessed knowledge about breast cancer. For example, one item assessed whether the participant knew if most lumps turn out to be breast cancer. (3) *Breast cancer treatment knowledge*: three items assessed knowledge about breast cancer treatment. For example, one item asked whether the participant knew if breast cancer had a good chance of being cured if it is detected early. Response options were *yes, no, and not sure* and correct answers were assigned 1 point, while incorrect and "don't know" answers were scored as 0. The total score was the mean of the scale scores. In *Holt et al. (2003)* test–retest reliability for this scale was acceptable, ranging from: $r = .45$–$.68$,
$p < .01$. The distributions of the scale scores and the total point score were acceptable with no skewness and kurtosis problems. The original scale was provided directly by the authors, and the translated measure used in the present study is available upon request.

### Attitudes toward breast cancer treatment

Attitudes toward breast cancer treatment were measured by asking patients to express the extent to which they thought that "regularly following breast cancer treatment regimens would be..." fundamental, unpleasant, useless, worrying, right, and reassuring. Each item ranged on a 5-point scale, from 1 (not at all) to 5 (completely). The three negatively keyed items were "reverse-scored." The measure was developed by the authors, following the recommendations of *Ajzen (1991)* for attitude measure development. The face validity of the scale as well as the clarity of the items, were preliminarily evaluated through a think-aloud procedure[1] in a sample of 25 volunteer breast cancers patients aged between 20 and 60 (mean = 44.04 years, SD = 9.44). The results confirmed thorough coverage of the intended theoretical construct with the set of items included in this measure. Furthermore, a pilot study (*Chirico et al., 2012*) revealed good internal consistency for this scale (Cronbach's alpha = 0.75) on a sample of 100 newly-diagnosed breast cancer patients aged 26–74 years  (mean = 45.22, SD = 9.27). Participants from the pilot study and the present study were both recruited from the same hospital, and in both instances, the measure was completed during their hospitalization. Internal consistency reliability was comparable in the current study (Cronbach's alpha = 0.72).

This measure was scored by calculating a mean score, with higher scores indicating more positive patient attitudes towards breast cancer treatment.

### Self-efficacy for behaviours related to coping with cancer

The Cancer Behaviour Inventory (CBI, version-2) (*Merluzzi et al., 2001*) is a 33-item questionnaire that assesses self-efficacy for coping with cancer and includes the following scales: (a) maintenance of activity and independence; (b) seeking and understanding medical information; (c) stress management; (d) coping with treatment-related side-effects; (e) accepting cancer/maintaining positive attitude; (f) affective regulation; and (g) seeking support. All items were rated on a nine-point Likert scale ranging from 1 (not at all capable) to 9 (completely capable). As described in Table 2, all of the sub-scales had an acceptable reliability (Cronbach's alpha ranging from 0.66 to 0.80) with the exception of affective regulation (Cronbach's alpha = 0.42), which was excluded from data analysis. For each subscale item scores were averaged, thus higher scores indicated more self-efficacy in each specific domains. A total self-efficacy score was also computed averaging these subscales' mean scores.

### Perceived social support

Perceived social support was measured by the Multidimensional Scale of Perceived Social Support (MSPSS) (*Zimet et al., 1990*), a 12-item questionnaire that measured the perceived adequacy of support given by three different sources: family (four items), friends (four items), and significant other persons (four items). All items were rated on a seven-point Likert scale ranging from 0 (strongly disagree) to 6 (strongly agree). All the scales had

[1] In "think aloud" interviews, the respondent is asked to think aloud as they answer questions thus verbalizing the thoughts that would normally remain silent during the process. Participants are not asked to explain or justify what they are doing and they are not asked to report their strategies. Thought processes are then examined for comprehension, recall and judgement difficulties. The methodology can be useful in identifying the face validity of the measure and any problematic questions (*Drennan, 2003*).

Chirico et al. (2015), PeerJ, DOI 10.7717/peerj.1107

**Table 2** Correlation matrix, descriptive statistics and reliability of the key variables of the study.

| | 1 | 2 | 2.1 | 2.2 | 3 | 3.1 | 3.2 | 3.3 | 4 | 4.1 | 4.2 | 4.3 | 4.4 | 4.5 | 4.6 | 4.7 | 5 | 5.1 | 5.2 | 5.3 | 6 |
|---|---|---|---|---|---|---|---|---|---|---|---|---|---|---|---|---|---|---|---|---|---|
| 1. Age | – | | | | | | | | | | | | | | | | | | | | |
| 2. Psychological distress | −.10 | – | | | | | | | | | | | | | | | | | | | |
| 2.1 Anxiety (STAI mean score) | .12 | .91 | – | | | | | | | | | | | | | | | | | | |
| 2.2 Depressive (CES-D mean score) | .06 | .92 | .69 | – | | | | | | | | | | | | | | | | | |
| 3. Perceived social support | −.05 | −.13 | −.10 | −.13 | – | | | | | | | | | | | | | | | | |
| 3.1 Family perceived support | −.04 | −.06 | −.05 | −.06 | .86 | – | | | | | | | | | | | | | | | |
| 3.2 Friends perceived support | .00 | −.08 | −.05 | −.08 | .81 | .54 | – | | | | | | | | | | | | | | |
| 3.3 Significant other persons perceived support | −.11 | −.18 | −.15 | −18 | .82 | .62 | .43 | – | | | | | | | | | | | | | |
| 4. Self-efficacy | −.13 | −.53 | −.48 | −.51 | −.18 | .05 | .10 | .29 | – | | | | | | | | | | | | |
| 4.1 Maintenance of activity and independence | −.09 | −.34 | −.33 | −.31 | .06 | −.02 | .02 | .15 | .69 | – | | | | | | | | | | | |
| 4.2 Seeking and understanding medical information | −.09 | −.26 | −.26 | −.23 | .12 | .02 | .07 | .20 | .69 | .39 | – | | | | | | | | | | |
| 4.3 Stress management | −.13 | −.60 | −.56 | −.56 | .11 | .03 | .04 | .22 | .80 | .47 | .47 | – | | | | | | | | | |
| 4.4 Coping with treatment-related side effects | −.07 | −.47 | −.43 | −.45 | .02 | −.06 | .01 | .11 | .79 | .40 | .44 | .68 | – | | | | | | | | |
| 4.5. Accepting cancer/maintaining positive attitude | −.09 | −.55 | −.50 | −.52 | .09 | .03 | −.04 | .24 | .82 | .62 | .40 | .70 | .66 | – | | | | | | | |
| 4.6. Affective regulation | −.01 | −.19 | −.16 | −.19 | .14 | .07 | .13 | .16 | .62 | .36 | .43 | .32 | .37 | .40 | – | | | | | | |
| 4.7 Seeking social support | −.14 | −.26 | −.17 | −.30 | .34 | .19 | .28 | .37 | .69 | .43 | .39 | .41 | .39 | .44 | .46 | – | | | | | |
| 5. Patients' knowledge | −.18 | −.22 | −.20 | −.21 | .07 | −.04 | .13 | .08 | .31 | .17 | .23 | .23 | .28 | .22 | .12 | .28 | – | | | | |
| 5.1 Mammography knowledge | −.10 | −.11 | −.10 | −.11 | .07 | .05 | .03 | .11 | .23 | .18 | .14 | .12 | .25 | .25 | −.01 | .20 | .76 | – | | | |
| 5.2 Breast cancer knowledge | −.17 | −.20 | −.18 | −.19 | .04 | −.06 | .10 | .05 | .21 | .10 | .22 | .16 | .15 | .12 | .14 | .18 | .78 | .30 | – | | |
| 5.3 Breast cancer treatment knowledge | −.11 | −.18 | −.16 | −.17 | .04 | −.09 | .18 | −.01 | .24 | .07 | .14 | .26 | .22 | .08 | .15 | .26 | .64 | .33 | .30 | – | |
| 6. Attitudes towards breast cancer treatment | −.09 | −.22 | −.17 | −.23 | .23 | .13 | .19 | .26 | .38 | .20 | .34 | .28 | .29 | .27 | .21 | .34 | .24 | .10 | .19 | .27 | – |
| Mean | 45.69 | 1.2 | 1.25 | 1.27 | 4.43 | 4.56 | 4.00 | 4.68 | 5.91 | 6.89 | 6.67 | 5.59 | 4.34 | 6.09 | 5.85 | 5.78 | 2.40 | 2.72 | 2.36 | 2.06 | 3.94 |
| SD | 10.01 | .46 | .47 | .54 | 1.11 | 1.28 | 1.45 | 1.32 | 1.28 | 1.46 | 1.82 | 1.88 | 2.10 | 1.64 | 1.27 | 2.00 | .86 | 1.27 | 1.41 | .82 | .71 |
| Cronbach's alpha | | | .87 | .88 | | .89 | .91 | .91 | | .78 | .68 | .70 | .80 | .76 | .42 | .66 | | | | | .72 |

**Notes.**

All the correlation coefficients are statistically significant at least at a *p*-level of .05, with the exception of underlined coefficients. In bold are reported the correlation between the main key variables of the study.

good reliability (Cronbach's alpha = 0.89, 0.91, 0.91, for family, friends and other persons, respectively). For each subscale a mean score was calculated based on the item scores, with higher scores indicating more perceived support. A total social support score was also calculated by averaging the scores of each subscale.

### Psychological distress

Anxiety was measured by the state form of the State-Trait Anxiety Inventory (STAI Form Y; *Spielberger, 1983*). Items like "I feel nervous" were rated on a four-point Likert scale ranging from 0 (almost never) to 3 (almost always). The scale showed a good reliability based on the data in this study (Cronbach's alpha = 0.87). Depression was measured by the CES-D scale (*Radloff, 1977*), a 20-item self-report scale designed to measure depressive symptoms in the general population but also used in cancer patients (*Van Wilgen et al., 2006*). Participants were asked to indicate how often, over the past week, they experienced each of the 20 symptoms described in the CES-D scale. Responses were made on a four-point scale ranging from 0 (rarely or not at all) to 3 (most of or all the time). The scale had a good reliability (Cronbach's alpha = 0.88) based on the data in this study. Item scores for each measure (i.e., STAI & CESD) were averaged to form an anxiety and a depression score with higher values indicating more anxiety and/or depression.

## Ethical considerations

The ethics committee of the National Cancer Institute 'Giovanni Pascale' Foundation approved the study (n.29/11). Informed consent was obtained from all participants. Data were confidentially gathered and collected anonymously with a smart code used to refer to the case. The voluntary nature of the study was emphasized and the authors have no conflicts of interest to report in the conduct of this study.

## Data analysis

Preliminarily, we verified that none of the main key measures in the model was correlated with tumour stage or familial history of breast cancer. Furthermore, the bivariate correlations between all the key measures used in the study and their descriptive statistics were calculated and presented in Table 2.

In order to test the hypothesized model, we used a Structural Equation Modelling (SEM) procedure. In particular we tested a mediational model, which hypothesized that age would be directly related knowledge that, in turn, would relate to attitudes and self-efficacy. Finally, the model also posed that these variables (i.e., attitudes and self-efficacy) and perceived social support would be directly related psychological distress. The direct and indirect effects of age and of knowledge on psychological distress were also evaluated. Finally, the direct effects of age on other variables (i.e., attitudes, self-efficacy, social support and psychological distress) were also estimated in order control for the effects of age on the hypothesized relationships in the model.

The model's parameters were estimated using the Maximum Likelihood (ML) estimation method through MPLUS-7 software (*Muthen & Muthen, 2012*). In the tested model, both STAI and CES-D scores were used as indicators of the latent variable

representing psychological distress; the three sub-scales of the MSPSS (i.e., family, friends and significant other person) were indicators of the latent variable social support; and the knowledge scales scores (i.e., knowledge about mammography, breast cancer, breast cancer treatment and early diagnosis of breast cancer) were used as indicators of the latent variable knowledge. For the latent variable of "self-efficacy," all the CBI subscales were considered as indicators with the exception of the "affective regulation" and the "social support" subscales. The former was excluded for its low reliability, the latter was excluded because the conceptual overlap and its high multicollinearity with MSPSS scales (details of the full measurement model can be obtained from the first author upon request). For the latent variable defined as "attitudes toward breast cancer treatment," an item-parcelling procedure was used (*Kim & Hagtevt, 2003*) in which the six items of the attitudes scale were randomly grouped and averaged yielding three separate parcels, which constituted three indicators of attitudes.

In order to evaluate the adequacy of the SEM analysis, we considered a variety of indices of the degree of fit between input data and model-based estimates. The literature indicates the following as good model-fit indices: TLI (Tucker-Lewis Index) or CFI (Comparative Fit Index) values close to 0.95; RMSEA (Root Mean Square Error of Approximation) value below 0.06 (*Hu & Bentler, 1999*), a $\chi^2$/df ratio below two (*Tabachnick & Fidell, 2007*). Finally, in order to analyze the indirect effects hypothesized, a SEM with a bias corrected (BC) bootstrap method was used to establish confidence intervals (CIs) for the indirect effects and confirm their statistical significance (*Preacher & Hayes, 2008*). In the present study, 95% confidence intervals were obtained with 1000 bootstrap resampling (*Preacher & Hayes, 2008*).

## RESULTS

Table 2 contains the bivariate correlations between the variables in the study. More specifically, the correlations between key constructs are presented in bolded text (i.e., between psychological distress, perceived social support, self-efficacy, knowledge and attitudes). The age of the patients correlated significantly only with their knowledge ($r = -.18$), which was correlated positively with attitudes ($r = .24$) and self-efficacy ($r = .31$); attitudes also correlated positively with social support ($r = .23$) and with self-efficacy ($r = .38$). The patients' knowledge ($r = -.22$), attitudes ($r = -.22$) and self-efficacy ($r = -.53$) were negative correlated with distress. Finally social support correlated negatively with self-efficacy ($r = -.18$).

As for the SEM analysis, which was performed to examine the mechanisms underlying and mediating the relationship between patients' age and psychological distress, the hypothesized model yielded very good fit indices (Chi-square $_{(106)}$ = 122.115; $\chi^2$/df = 1.15; CFI = 0.98, RMSEA = 0.034, SRMR = 0.058), in line with the criteria reported above. The measurement parameters of the model's latent constructs were statistically significant (all loadings >0.51). Figure 1 shows the latent path estimations and latent covariance estimations.

As reported in Fig. 1 the patients' age was negatively associated with the patient's knowledge ($\beta = -0.22$), which, in turn, was positively related to both attitudes toward breast cancer treatment ($\beta = 0.39$) and coping self-efficacy ($\beta = 0.36$). Self-efficacy, in turn, represented the only variable of the hypothesized model that was directly and negatively related to patients' psychological distress ($\beta = -0.68$), which, contrary to hypothesis, is not significantly related to either attitudes or social support. Finally, the analysis of the indirect effects of the hypothesized model revealed a significant indirect effect of knowledge on psychological distress ($\beta = -.25$; BC bootstrap CIs: from $-.42$ to $-.08$) through the mediation of the self-efficacy. Overall, the tested model accounted for about 50% of the variance of the patients' psychological distress.

## DISCUSSION

The main aim of the present study was to gain a more thorough understanding of the contribution of critical variables that determine individual differences in the level of psychological distress experienced by newly diagnosed breast cancer patients before they begin treatment. Thus, variables that could exacerbate or lessen patients' psychological distress (e.g., *Grassi et al., 1993*; *Merluzzi & Sanchez, 1997*; *Merluzzi et al., 2001*; *Gilbar, 2003*; *Mosher & Danoff-Burg, 2005*; *Friedman et al., 2006*; *Arora et al., 2007*; *Henselmans et al., 2010*; *Heitzmann et al., 2011*; *Philip et al., 2013*), were included in a model linking age, knowledge about breast cancer, attitudes toward cancer, coping efficacy, social support and distress outcomes. The findings of a SEM analysis substantially confirmed our hypothesized path model.

First of all, age was related to patients' level of knowledge about breast cancer, specifically, older the patients had less knowledge about breast cancer and its treatment. This result is not consistent with those from other studies (e.g., *Lukwago et al., 2003*) and other populations (e.g., African Americans) suggesting that patients' knowledge could differently change with age as a function of the specific cultural context or at a latter phase in the treatment stage of cancer. In this study, older patients with lower knowledge, in turn, show lower scores in coping self-efficacy and a higher level of psychological distress. These results are in line with social cognitive theory (*Bandura, 1997*), which posits that patient's knowledge can directly relate to self-efficacy. Finally, in the next phase of the model, our results confirmed past research showing that self-efficacy mitigated psychological distress in cancer patients (e.g., *Howsepian & Merluzzi, 2009*; *Heitzmann et al., 2011*; *Philip et al., 2013*) and in particular in breast cancer patients (*Henselmans et al., 2010*).

In contrast, our younger patients showed more knowledge and positive attitudes toward breast cancer treatment, perceived themselves as more efficacious in coping with their cancer condition, and were less distressed. Thus, early on in the cancer trajectory, age can be considered as a crucial precursor of patients' distress based mainly on deficits in knowledge, which then leads to lack of confidence in coping efficacy and distress. This sequelae of effects could be contrasted with past literature, which showed that younger age was related to greater distress (e.g.,*Van't Spijker, Trijsburg & Duivenvoorden, 1997*; *Avis et al., 2012*; *Mertz et al., 2012*). However, this could be explained hypothesizing

that the relationship between age and distress is strictly dependent by the influence of a third variable, namely the level of knowledge, which is strictly dependent by the cultural context. Consistently, some scholars (*Grassi et al., 2015*) suggested the need to take into consideration the putative effects of variables that are strictly related to the cultural context in which the study is performed. Thus, future research might contrast age and knowledge in several different cultural contexts including Italy to determine if there are important cultural differences in age and knowledge about breast cancer.

In our results, the lack of a significant effect of social support on distress was unexpected. Generally, the research on social support confirms its positive influence on outcomes such as distress; however, its role may be related to where people are on the cancer trajectory. For example, *Philip et al. (2013)* found that social support was not as important as coping self-efficacy for survivors. Perhaps this is also the case for patients who are post-diagnosis but pre-treatment. That is, the patients in this study are at a point in the cancer trajectory where they may still rely on their own coping efficacy and social support has not yet been engaged as a key variable in their perceived ability to cope with the disease. As they progress into active treatment (surgery and adjuvant chemotherapy), the role of social support may emerge as a critical component of the coping process.

There were also a no significant direct effects of attitudes on patients' distress, but there is a negative bivariate correlation between the two. Furthermore, there is a positive relationship between attitudes and self-efficacy. These results suggest that attitudes may be operating through self-efficacy to augment the mitigation of distress. It is likely that, because attitudes are able to directly predict volitional and goal oriented health behaviors, they are less related to mood state (i.e., *Manning & Bettencourt, 2011*) than to agency, which is reflected in its relationship with coping efficacy.

The abatement of distress early in the diagnosis and treatment of cancer may have long-term beneficial effects. In her bio-behavioral model (*Andersen et al., 2008*), Andersen, stated that an important sequela of distress is (non)compliance. Many studies have shown a positive association between distress and decreased acceptance of and compliance with treatment (*Ayres et al., 1994*; *Colleoni et al., 2000*; *Bui et al., 2005*), which may, in turn, affect disease outcomes, the prevention of recurrence, and long-term survivorship. Knowing some of the risk factors that were present in this study may help guide a process of early intervention to avoid the exacerbation of distress. According to our findings, it might be crucial for interventions to focus on patients' beliefs and knowledge about breast cancer and its treatment. In fact, according to our findings and other scholars' suggestions (*Heitzmann et al., 2011*; *Yi & Park, 2012*), an increase in patients' knowledge about cancer, its detection, and its treatment can directly improve self-efficacy to cope with several aspects of their illness and, indirectly, their psychological distress (*Chen et al., 2008*). In addition, there are several studies that have recently investigated the effects of psychosocial interventions that enhance self-efficacy, showing a reduction of patients' psychological distress for both professionally run programs (e.g., *Abernethy et al., 2010*; *Smith et al., 2011*) as well as those conducted by trained breast cancer survivors (*Yi & Park, 2012*).

The use of a cross-sectional design is one of the limitations of the present study; thus, future studies should explore some of the same issues but in a longitudinal design. Furthermore, it is important to reiterate that our findings stemmed from a sample of newly diagnosed patients. Future studies might explore the social cognitive mediation model proposed in the present investigation to account for possible indicators of psychological distress in patients at different stages of their illness and its treatment. Finally, as mentioned previously, some specific findings (i.e., the relationship between age and patients' knowledge) might be related to the specific cultural context in which the study was performed; thus, the comparative culturally-based study of processes early in the cancer trajectory would help determine those aspects that are universal and those that have more cultural specificity.

## CONCLUSION

The findings of the present research have direct, practical implications for healthcare professionals who work with breast cancer patients. They can play a crucial role in assessing and imparting correct knowledge about breast cancer and its treatment shortly after diagnosis. In fact, consistent with our findings, increasing patients' knowledge about breast cancer could directly improve their self-efficacy to cope with cancer and psychological distress.

### Funding
The authors declare there was no funding for this work.

### Competing Interests
The authors declare there are no competing interests.

### Author Contributions
- Andrea Chirico conceived and designed the experiments, performed the experiments, analyzed the data, wrote the paper.
- Fabio Lucidi conceived and designed the experiments, analyzed the data, contributed reagents/materials/analysis tools, wrote the paper, reviewed drafts of the paper.
- Luca Mallia analyzed the data, wrote the paper, prepared figures and/or tables.
- Massimiliano D'Aiuto wrote the paper.
- Thomas V. Merluzzi wrote the paper, reviewed drafts of the paper.

### Human Ethics
The following information was supplied relating to ethical approvals (i.e., approving body and any reference numbers):

The ethics committee of the National Cancer Institute 'Giovanni Pascale' Foundation approved the study through formal authorization of the scientific board protocol number 29/11. Informed consent was obtained from all participants. Data were confidentially

gathered and collected anonymously via a smart code to refer to the case, and the voluntary nature of the study was emphasized.

## Supplemental Information

Supplemental information for this article can be found online at http://dx.doi.org/10.7717/peerj.1107#supplemental-information.

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
