# Peer review of "Indicators of distress in newly diagnosed breast cancer patients"

_PeerJ, doi:10.7717/peerj.1107_

## Round 0.1 · original submission · Major Revisions

· Academic Editor

Major Revisions

Please review all comments raised by reviewers and provide a rebuttal letter that contains a point-by-point response for each reviewer that discusses how and where the issue was addressed in the revised manuscript.

·

Basic reporting

Overall, this is an excellent article. It is timely, and recommend publication.

Experimental design

The design of the study was simplistic in gathering responses to the items which enabled to authors to make multiple correlations. At the end of the 'Discussion', the authors state this is a cross-sectional design (ie this should be stated in line 134-135.

The design of the study was also complex as the authors attempted to design and validate (with empirical data) a model of psychological distress and its mediators.

The sentence from 223-227 should be cleaned up, or broken into two sentences as far too much for one sentence.

271-286 is inconsistent with the background; the authors should correlate their findings to other published works.

Validity of the findings

The authors did an excellent job in explaining each instrument with inclusion of reliability adn validity data. They did fall short in the instrument "Attitudes toward breast cancer treatment" which is a instrument designed by the authors, but lacks any reliability or validity data. At the very least, these authors could have obtained reliabilty measures of 'face value', 'professional scoring', and test-retest. In addition, small measures of validity could examine he relationship between these items and other instruments' items.
These results could have been published prior to this manuscript and would validate the ENTIRE process that is explained in this paper.

Additional comments

This is an excellent article, although have a couple of recommendations to strengthen the article..
1. Commend you on confining the study to cancer type, and identifying the timeline. Too many studies are conflicted due to inclusion of multiple cancers at different points on the cancer trajectory.
2. Due to differences between Italy and other countries, incl. USA, would explain the practice of Italian surgeons, ie 'hospitalized patients'. i.e. are they just 'hospitalized' for their outpatient surgery, OR were all of these patients having larger procedures, ie mastectomy, axillary node dissection, or reconstruction. In USA, lumpectomy with sentinel lymph node is a same day surgery wtihout admission to hospital. Were any of these patients postoperative? or were any of them preoperative? It might be helpful to add a few demographics about their breast cancer, stage, etc. This factor alone could explain a time period of increased distressed, ie worried before their surgery, vs. in pain postoperatively3. Also, would have been good to identify their level of distress (although did not directly measure it)
4. Since your measures are used to define your model, ABSOLUTELY need reliability and validity coefficients....this lapse is significant!

Reviewer 2 ·

Basic reporting

No comments

Experimental design

No comments

Validity of the findings

no comments

Additional comments

The present manuscript reports on a cross sectional study of predictors of distress among in patient breast cancer survivors immediately after their diagnosis. The authors used SEM to evaluate the relationships among patient age, knowledge of cancer management and treatment, attitudes toward cancer, social support, coping efficacy, and distress. Age was negatively associated with knowledge and knowledge was associated with attitudes and coping self-efficacy. Only coping self-efficacy was associated (negatively) with distress.

The manuscript has several strengths. In general it is well-written, although it is a bit too long. It has an interesting sample, in-patient new diagnosed breast cancer patients, and it includes a theoretically based model that includes age and cancer knowledge as predictors. However, the manuscript in its present form however, has several weaknesses, described below, which limit its appeal.

1.) The present manuscript would benefit from more discussion of previous longitudinal studies (e.g. Carver et al. 1993, JPSP, Stanton and Snider 1993 Health Psychology) to evaluate optimism, coping and other predictors of distress in the time surrounding breast cancer diagnosis. I believe this paper has something to add to the literature, the authors just need to clarify how their study compares and adds to the earlier work.

2.) The introduction. The introduction is way too long. It needs to be much more concise. It also needs many more citations for the various claims it makes.

3.) In general, the manuscript is well written but there are a few rough places. Lines 39 through 56 would benefit from some editing for English language.

4.) The paragraph in lines 62-72 seem to add anything. It feels redundant and is an example of where the intro feels long.

5.) Line 96 social support “might” act as a protective factor? It is this reviewers understanding that social support is a well-known buffer of the effects of stress. As such, it should be added to the model to see how it fits w/ the other variables, but it is not a question of if, but more some how it fits in the model. This paragraph should reflect that.

6.)Line 115-118 was good!

7.) It is recommended that the intro be more clearly organized around the critical variables listed in line 124, age, knowledge, attitudes, social support, self-efficacy, with a brief paragraph for each. As it is now, there is a lot of discussion about age and social support but much less about knowledge and attitudes. If that is because there is less known about knowledge and attitudes, this is a gap that needs to be illuminated in the text.

8.) The introduction needs to make it clear that the study is cross sectional, not longitudinal. This reviewer was surprised in the Methods paragraph at Line 143 to learn that the study was cross sectional.

9.) Method needs to include how demographic (e.g. age) and treatment (e.g. stage) variables were measured. Where these data collected during the interview or were they obtained from a medical record review?

10.) Information about stage of cancer at diagnosis should be included. Also surgery type (e.g., lumpectomy, mastectomy)

11.) The attitudes toward breast cancer treatment measure is a new and unpublished measure. As such, more information about its creations and psychometric properties should be included here.

12.) More information is needed regarding scoring of the self-efficacy scale (Cancer Behavior Inventory) and subscales.

13.) More information is needed regarding scoring of the perceived social support measure, (Multidimensional Scale of Perceived Social Support) and subscales.

14.) Please clarify whether you used the state or trait anxiety scale.

15.) The analysis section is well done. The results section is well-written.

16.) Line 261 – please speculate in the discussion why the attitudes and social support were not significantly related to distress.

17.) Line 278. Younger patients were less distressed (and had more positive attitudes and greater coping efficacy). This seems different than most of the literature which reports younger age is usually associated with greater distress. This seems important. If younger age is associated with and more knowledge better coping efficacy and that mean less distress. This is an important previously unreported (I think) association.

18.) Line 290 the sentence here seems random – it doesn’t really fit.

19.) In general, the discussion is too long and missing citations just like the intro. The section in paragraph 316 is good, but the whole discussion needs to be more concise.

·

Basic reporting

In this study women with newly diagnosed breastcancer completed questionnaires concerning their knowledge about breast cancer treatment, coping efficacy and distress.
It was found that older women had less knowledge about breast cancer treatment and that less knowledge was associated with diminished coping, which led to increased distress,
Included women varied in age from 20 to 74, and were interviewed by a psychologist during the hospital admittance for the first surgery.
Distress and coping are important features in breast cancer treatment and this research is an interesting starting point concerning distress and its relation to treatment compliance and treatment result.
However I do have several questions and concerns:
First of all the age distribution of the included patients is not given. There is a mean age given but the number of women post-menopausal for instance is not given. Other factors such as educational level, socio-economic status, positive family history etc that may influence knowledge and expectations about breast cancer are not mentioned.

Experimental design

The fact that age has a negative influence on knowledge is not new, however the consequences thereof may be difficult to interpret. Interviewing women while they are admitted to the hospital may have a negative influence on experienced distress. In addition, I would expect that these women have received enough information concerning the treatment considering the fact that they are awaiting surgery. The authors do not mention whether or not in this group patients have received neo-adjuvant treatment for their breast cancer nor is the stage of the breast cancer mentioned. These facts may have additional impact on perceived stress
If you want to examine the relationship between knowledge and coping you need to correct for age since age in itself leads to worsening of coping. It is not mentioned in the statistical analyses if this is done

Validity of the findings

Based on the results the authors come to the conclusion that ‘ these findings establish early indicators of distress that may affect compliance with treatment and quality of life but are also amenable to change’. Both these conclusions are not supported by the presented data. QoL is not measured and compliance with treatment is not assessed in these patients.
I would be very careful with these statements

The study ( and thus the manuscript) would benefit from a longitudinal follow up in order to find out if distress indeed has the aforementioned consequences.

Additional comments

I would consider this study a good starting point for additional follow-up studies. Considering the fact that data are limited to one point in time (and a very stressful point at that) i would be very careful with predictions of the consequences.

---

## Round 0.2 · Minor Revisions

· Academic Editor

Minor Revisions

Although most issues have been addressed, the psychometric properties issues have not been fully addressed. Given the importance of this issue, the authors need to provide further information to fully resolve this.

·

Basic reporting

This is an excellent article. We asked the authors to describe preliminary psychometric testing of their instrument. Results are reported, atlhough never clearly explained. Is this data published elsewhere that can be used as a 'refer to'?

Otherwise, in the first segment where they talk about the r scores, they need to describe the sample, methods, etc. and how this was derived. IDEALLY, this would be published ahead of this article if there is any way to do it.

Experimental design

no problem

Validity of the findings

no problem except in providing preliminary psychometric information about their instrument -- see above

Additional comments

EXCELLENT article -- I enjoyed reading it. Just need to clean up one last section and that refers to your instrument.....j

---

## Round 0.3 · accepted · Accept

· Academic Editor

Accept

All issues have now been fully addressed by the authors.

·

Basic reporting

yes

Experimental design

Improved from previous version

Validity of the findings

Much better with revisions